# Predictors of Metabolic Syndrome in Adults and Older Adults from Amazonas, Brazil

**DOI:** 10.3390/ijerph18031303

**Published:** 2021-02-01

**Authors:** Élvio Rúbio Gouveia, Bruna R. Gouveia, Adilson Marques, Miguel Peralta, Cíntia França, Alex Lima, Alderlane Campos, Jefferson Jurema, Matthias Kliegel, Andreas Ihle

**Affiliations:** 1Department of Physical Education and Sport, University of Madeira, 9020-105 Funchal, Portugal; cintiarnf@gmail.com; 2LARSYS, Interactive Technologies Institute, 9020-105 Funchal, Portugal; bgouveia@esesjcluny.pt; 3Center for the Interdisciplinary Study of Gerontology and Vulnerability, University of Geneva, 1205 Geneva, Switzerland; matthias.kliegel@unige.ch (M.K.); andreas.ihle@unige.ch (A.I.); 4Regional Directorate of Health, Secretary of Health of the Autonomous Region of Madeira, 9004-515 Funchal, Portugal; 5Saint Joseph of Cluny Higher School of Nursing, 9050-535 Funchal, Portugal; 6CIPER, Faculdade de Motricidade Humana, Universidade de Lisboa, 1495-751 Lisbon, Portugal; amarques@fmh.ulisboa.pt (A.M.); mperalta@fmh.ulisboa.pt (M.P.); profalexbarreto@hotmail.com (A.L.); 7ISAMB, Faculdade de Medicina, Universidade de Lisboa, 1649-020 Lisbon, Portugal; 8Centro Universitário do Norte, 69020-010 Manaus, Brazil; 9Secretaria Municipal de Educação, 69055-010 Manaus, Brazil; alderlaneoliveira@hotmail.com; 10Amazonas State University, 69065-001 Manaus, Brazil; jjurema@uea.edu.br; 11Department of Psychology, University of Geneva, 1205 Geneva, Switzerland; 12Swiss National Centre of Competence in Research LIVES—Overcoming Vulnerability: Life Course Perspectives, Lausanne and Geneva, 1022 Chavannes-près-Renens, Switzerland

**Keywords:** cardiometabolic risk factors, health status, vulnerability, sex-related differences, public health, Amazona—Brazil

## Abstract

Metabolic syndrome has been considered a factor of vulnerability and a major public health problem because it increases the risk of cardiovascular disease and type 2 diabetes. The present study from Amazonas, Brazil aimed to estimate the prevalence of the individual and general components of metabolic syndrome in adults and older adults and identify the independent predictors of metabolic syndrome. The sample of the present cross-sectional study comprised 942 participants (590 women), with a mean age of 59.8 ± 19.7 (range: 17.5 to 91.8). Blood pressure in men (62.5%), abdominal obesity in women (67.3%), and lower high-density lipoprotein cholesterol (HDL-C) in both (52.2% in men and 65.0% in women) were the most prevalent individual risk factors for metabolic syndrome. Women had a higher prevalence of abdominal obesity (*p* < 0.001), low HDL-C (*p* < 0.001), and metabolic syndrome (*p* < 0.001) than men; however, opposite results were seen in men for blood pressure (*p* < 0.001). The overall prevalence of metabolic syndrome was 47.5%. Advanced age, being female, having a higher body mass index, and a having lower educational level independently increased the odds of metabolic syndrome. Due to the association of metabolic syndrome with deterioration of health status and increased vulnerability, this study sustains the need for early public health interventions in the Amazonas region.

## 1. Introduction

Metabolic syndrome is a clustering of interrelated cardiometabolic risk factors, which more often occur together than individually [1]. The metabolic abnormalities that characterize metabolic syndrome include increased blood pressure, elevated blood glucose, excess body fat around the waist, and abnormal cholesterol or triglyceride levels [2].

The current literature claims metabolic syndrome as a factor of vulnerability [3,4,5] and a major public health problem, the prevalence of which is increasing worldwide [6,7,8,9]. Individuals with metabolic syndrome are 2.5 times more likely to die from cardiovascular disease compared with their peers with no metabolic syndrome [10]. Additionally, these individuals are five times more vulnerable to develop type 2 diabetes mellitus [11].

In the last decade, several studies have contributed to estimating the prevalence of metabolic syndrome and its risk factors. This mapping has vital importance in early risk identification and increases the effectiveness of specific interventions aimed to prevent future deterioration of the health status. In Brazil, a representative sample analysis (National Health Survey in 2013) showed a prevalence of metabolic syndrome of 7.5% in men and 10.3% in women [12]. de Carvalho Vidigal [13] in a systematic review including healthy Brazilian adults aged between 19 and 64 years old, reported a prevalence of 29.6% for metabolic syndrome. In this review, 10 studies were considered; however, none of them gathered information from the Amazonas region. The prevalence reported in this review for the context of Brazil context was higher than the average of the world’s adult population, which is estimated between 20 and 25% [11]. However, the higher prevalence was seen in Portugal (47.2%, similar in females and males; [14]), in Indonesia (28% in men and 46% in women), and in the Netherlands (36% in men and 24% in women; [15]). Our study, using a reliable methodology to diagnose metabolic syndrome, adds knowledge to this topic in the Amazonas region, where information is scarce. To the best of our knowledge, this is the first study in a large adult lifespan sample from this region of Brazil. This issue reaches greater importance in this particular region because it is characterized by socioeconomic and health inequalities, which place serious challenges on the health public system [16].

Among the modifiable explanatory factors for this phenomenon, the increase in obesity and the adoption of sedentary lifestyles during past decades has been highlighted [12,17,18]. Other factors have also been shown to play a role in the development of metabolic syndromes, such as individual variables (i.e., sex), genetic predisposition [19], smoking and unhealthy eating patterns [18], aging [13], and contextual involvement [12]. However, such relationships are not fully understood so far in the literature. For example, concerning sex-related differences, some studies have not found any differences between men and women [20,21,22,23]. Others have reported a higher prevalence of metabolic syndrome among men compared to women [15,24], and others found a higher prevalence in women compared with men [25,26,27]. These contradictory results can open new avenues in the research for possible explanations that go beyond the differences associated with the hormone levels commonly accepted [28].

There is a consensus in the literature that older age, a lower education level, low physical activity, and higher a body mass index increase the odds of metabolic syndrome [28]. First, the age-related increase in the prevalence of metabolic syndrome is significantly influenced by the large prevalence of metabolic risk factors developed at the oldest ages, in particular >65 years [29]. Second, the most disadvantaged people who had a lower education level, were unemployed, and of a lower income level had more vulnerability to develop metabolic syndrome [30]. Third, engagement in regular physical activity has been considered an important lifestyle intervention to prevent metabolic syndrome. Some studies have proved that regular physical activity reduces risk factors related to metabolic syndrome and associated vascular diseases [31]. However, the relationships between physical activity and metabolic syndrome seem to be dependent on factors related to the type and intensity of physical activity as well as individual variables and contextual involvement. Finally, the clustering of multiple risk components within metabolic syndrome is widely thought to occur as a result of obesity, specifically, abdominal obesity [32].

Information on the prevalence of metabolic syndrome and its relationship with independent predictors, such as demographics, lifestyle, and clinical measures, needs more clarification, in particular among the population of the state of Amazonas. Thus, the present study aimed to estimate the prevalence of the individual and general components of metabolic syndrome, considering sex-related differences, and assess the impact of age, sex, BMI, education level, and physical activity on the likelihood that participants present metabolic syndrome.

## 2. Methods

### 2.1. Sample and Study Design

The sample of the present cross-sectional study comprised 942 participants (352 men and 590 women), with a mean age of 59.8 ± 19.7 years (range: 17.5 to 91.8). From those, only 910 participants (569 women) fulfilled all the criteria for the diagnosis of metabolic syndrome. Data were obtained from the Health, Lifestyle, and Functional Fitness in the Adult and Older People from Amazonas, Brazil (SEVAAI) study. Data were collected between 2016 and 2017. Participants were volunteers recruited via advertisements distributed through newspapers, local radio, churches, schools, and senior centers. The inclusion criteria were living in one of three geographic areas of Brazil (Fonte Boa, Apuí, and Manaus), voluntary motivation to participate in the study, and being able to walk independently. The only exclusion criterion was the inability to understand and follow the assessment protocol of the study. For the sample size calculation, we used the GPower, (Heinrich-Heine-University, Düsseldorf, Germany; 3.1.9.7 software) [33]. A priori, logistic regression analysis indicated that a total sample size of 824 participants was needed to achieve 85% power and an odds ratio of 1.3 at the 0.05 level of significance. All participants gave informed consent, and the present study included adherence to the declaration of Helsinki and had been approved by the local ethics commission (CAAE: 56519616.6.0000.5016; Number: 1.599.258; Brazil Platform).

### 2.2. Instruments

#### 2.2.1. Clinical Analysis

Several clinical parameters were assessed by specialized clinical analysis laboratory technicians; in particular, glucose level, high- and low-density lipoproteins (HDL and LDL, respectively), and triglycerides. Blood was collected by venipuncture from all participants, in a fasted state (more than 8 h), between 7 am and 9 am. For this purpose, needles were used, as well as syringes and collection tubes. After collecting 7 mL of blood from the antecubital vein to a dry tube with silica gel, i.e., an accelerator of serum separation, the tubes were identified and transported to the laboratory. After an hour of rest, the serum was separated by centrifugation at room temperature for 15 min at 3500 rotations per minute.

#### 2.2.2. Blood Pressure

After 20 min of rest, blood pressure was assessed (OMRON M6 HEM-7001-E; Omron Corporation, Kyoto, Japan). Participants were evaluated in a seated position, with their right arm at the heart level, then asked to relax and stay still during the measurement. At least two measurements were made (separate repeated measurements every 1–2 min), taking the average value between both as long as the difference did not exceed 5 mm/Hg.

#### 2.2.3. Anthropometric Measurements

Waist circumference was measured halfway between the lower costal margin and the iliac crest in an orthostatic position to the nearest 0.1 cm, according to the International Society for the Advancement of Kinanthropometry protocol [34]. Body mass index was calculated from weight and height (kg/m^2^). Height was recorded accurately to 1 mm with a portable stadiometer (SECA 217, Hamburg, Germany). Body mass was measured on a balance-beam scale accurate to 0.1 kg (Seca Optima 760, Hamburg, Germany).

#### 2.2.4. Physical Activity

Physical activity was assessed in face-to-face interviews using the Baecke questionnaire [35]. This questionnaire includes a total of 16 questions classified into three specific domains: physical activity at work/household activities (HS), sport, and leisure time. Numerical coding for most response categories varied from 1 to 5 (Likert scale), ranging from never to always or very often. The questionnaire also provides a measure of total physical activity (PA), which is the sum of these three specific domains. Numerical coding for most response categories varied from 1 to 5 (Likert scale), ranging from never to always or very often.

#### 2.2.5. Socioeconomic Status

Socioeconomic status was assessed with the Brazilian Economic Classification Criterion developed by the Brazilian Association of Research Companies [36], considering each individual’s education level. To assess the education level, participants were asked to indicate their level of schooling. The following scores were considered for this purpose: 0 = illiterate/incomplete primary; 1 = complete primary/incomplete junior high; 2 = complete junior high/incomplete secondary; 3 = complete secondary/incomplete higher; 4 = complete higher.

#### 2.2.6. Determination of Metabolic Syndrome

The following criteria were defined for the diagnosis of metabolic syndrome, based on conventional cut-off values suggested by the consensus of the International Diabetes Federation Task Force on Epidemiology and Prevention; National Heart, Lung, and Blood Institute; American Heart Association; World Heart Federation; International Atherosclerosis Society; and International Association for the Study of Obesity (IDF/NHLBI/AHA/WHF/IAS/IASO [1]): (a) waist circumference ≥94 cm in men and ≥80 cm in women; (b) triglycerides ≥150 mg/dL (1.7 mmol/L); (c) cholesterol-HDL <40 mg/dL (1.0 mmol/L) in men and <50 mg/dL (1.3 mmol/L) in women; (d) systolic blood pressure ≥130 mmHg and/or diastolic ≥85 mmHg; and (e) fasting blood glucose ≥100 mg/dL. A coding of 1 was assigned to the presence of the respective risk factor and 0 to its absence. These codes were used for the final diagnosis of metabolic syndrome, with the presence of any of 3 out of the 5 listed risk factors constituting a diagnosis of metabolic syndrome [1].

### 2.3. Statistics

Based on the criteria of IDF/NHLBI/AHA/WHF/IAS/IASO [1], the prevalence of metabolic syndrome was expressed by a percentage. All data were tested for normality, and preliminary analyses were performed to ensure no violation of the assumptions. An independent-samples *t*-test was conducted to compare the means in the descriptive characteristics of the samples, according to the presence/absence of metabolic syndrome. In addition, to study the association between sex and categories of metabolic syndrome, the chi-squared test for independent measures was used. Finally, using direct logistic regression, several models were tested in order to identify what factors better explained the likelihood that participants would present metabolic syndrome. The final model contained age, sex (nonmodifiable variables), body mass index (BMI), education, and physical activity (modifiable variables). Odds ratios and 95% confidence intervals were also presented in order to quantify the change in odds of having metabolic syndrome when the value of a predictor increases by one unit. All statistical analyses were performed using IBM SPSS (IBM Corp., Armonk, NY, USA), version 26. The significance level was defined as *p* < 0.05.

## 3. Results

### 3.1. Descriptives

The means, standard deviations, and CI (95%) of the descriptive characteristics of the samples by sex are presented in Table 1. An independent-samples *t*-test was conducted to compare the descriptive characteristics for men and women. Men had significantly higher systolic and diastolic blood pressure, higher waist circumference, and higher physical activity than women. On the other hand, women had a significantly higher resting heart rate, higher fasting blood glucose, higher cholesterol-total, higher cholesterol-HDL, higher cholesterol-LDL, higher BMI, and a higher education level than men.

### 3.2. The Prevalence of Each Risk Factor for Metabolic Syndrome

Sex-related differences in the prevalence of each risk factor for metabolic syndrome are presented in Table 2. Blood pressure (62.5%) in men and abdominal obesity (67.3%) in women were the most prevalent risk factors of metabolic syndrome in this sample. Additionally, a low value of high-density lipoproteins (HDL; 52.2% and 65.0%) was the second risk factor with a higher prevalence in this sample in men and women, respectively.

The total prevalence of metabolic syndrome in this sample was 47.5% (39.6% in men and 52.2% in women). The chi-squared for independent measures showed higher prevalence in women in abdominal obesity [χ^2^ (1, *n* = 942) = 80.94; *p* < 0.001, phi = 0.29], low HDL [χ^2^ (1, *n* = 910) = 14.64; *p* < 0.001, phi = 0.13], and metabolic syndrome [χ^2^ (1, *n* = 910) = 13.59; *p* < 0.001, phi = 0.12]. However, men had elevated blood pressure [χ^2^ (1, *n* = 942) = 17.03; *p* < 0.001, phi = −0.13].

### 3.3. Predicting the Likelihood of Presenting Metabolic Syndrome

Logistic regression was performed to assess the role of age, sex, BMI, education level, and physical activity for the likelihood that participants would present metabolic syndrome. The full model containing all predictors was statistically significant (X^2^ (5, *n* = 906) = 198.34, *p* < 0.001), indicating that the model was able to distinguish between the presence/absence of metabolic syndrome. The model as a whole explained between 19.7% (Cox and Snell R square) and 26.2% (Nagelkerke R square) of the variance in metabolic syndrome status and correctly classified 71% of the cases. As shown in Table 3, with one exception (physical activity), all predictors made a unique statistically significant contribution to the model. The strongest predictor of metabolic syndrome in this sample was sex, recording an odds ratio of 1.67. This suggests that for women, the odds of presenting metabolic syndrome increases by a factor of 1.7. The second stronger predictor of metabolic syndrome was BMI (OR: 1.18, *p* < 0.001). This suggests that for each 1 kg/m^−2^ increase in BMI, the odds of presenting metabolic syndrome increases by a factor of 1.2. Interestingly, education was also a significant predictor, with an odds ratio of 1.12. This indicates that participants with low education levels were more likely to present the metabolic syndrome. For every cycle increased level of education, the odds of presenting metabolic syndrome decreased by a factor of 1.12. Finally, age (OR: 1.03, *p* < 0.001) was also a significant predictor of metabolic syndrome. This suggests that for each one-year increase in age, the odds of presenting metabolic syndrome increases by a factor of 1.03.

## 4. Discussion

This cross-sectional study sought to identify the most prevalent risk factors for metabolic syndrome and assess the relative impact of age, sex, BMI, education level, and physical activity on the likelihood that participants present metabolic syndrome. Our results showed that blood pressure in men, abdominal obesity in women, and lower HDL-C in both were the most prevalent individual risk factors for metabolic syndrome. Importantly, women had a higher prevalence of abdominal obesity, low HDL-C, and metabolic syndrome than men. On the other hand, men showed more elevated blood pressure than women. Finally, our results showed that advancing age, sex (female), having a higher BMI, and a lower educational level independently increase the odds of metabolic syndrome.

First, our results support that men have more elevated blood pressure levels in comparison to women. Similar results were seen in India [37], Korea [38], and the United States [39]. However, recently, opposite results were seen in Bangladesh, showing a higher prevalence of hypertension among women [40]. Several sex-related determinants of hypertension (i.e., age, dietetic habits, education, BMI, glucose) have been identified as important moderators able to explain those differences. The more elevated blood pressure levels found in our study, in particular in men, should trigger concrete nonpharmacological intervention actions at the community level. Among nonpharmacological interventions, there are important lifestyle modifications such as a healthy diet, weight loss, a reduction of sodium and alcohol intake, an increase of potassium intake, and an increase of weekly physical activity (in particular, aerobic exercise, dynamic resistance training, and isometric resistance training) [41].

Regarding central obesity, our results corroborate that women have a higher prevalence of abdominal obesity than men [42]. This might be explained by the more favorable fat distribution in women. Since abdominal obesity is highly correlated with metabolic diseases, efforts to reduce or prevent the deposition of intra-abdominal body fat might serve to reduce or prevent metabolic syndrome, particularly in women.

Considering all the five risk factors for metabolic syndrome, the second most prevalent in both men and women was the low concentration of HDL-C, which seems to be protective in the development of cardiovascular diseases [43]. The prevalence of concentrations of HDL-C levels above the cut-off points in women is significantly higher than in men, which corroborates the results achieved in other studies [44]. This finding could be explained in part by changes in their hormonally mediated cholesterol metabolism after menopause [31,45].

The prevalence of metabolic syndrome found in our study is very high (47.5%). The National Health Survey (2013) reported much lower metabolic syndrome prevalence (8.5%) in a sample of 59.402 Brazilians, being significantly higher in women than in men. Other studies developed in Iran [8] (30.1% in men and 55% in women) and Mexico [9] (48.9% in men and 60.4% in women) showed a higher prevalence of metabolic syndrome than our results and confirmed the tendency of women to present higher values compared with men, as we found in our study. In the context of Brazil, the closest results found in comparison with our study were presented by de Carvalho Vidigal et al. [13] According to that systematic review, the average prevalence of metabolic syndrome in Brazil was 29.6%. Even so, the results are much lower than ours. This can be explained in part by the socioeconomic and health inequalities that place serious challenges to the public health system [16] as well as the lower proportion of people practicing exercise in leisure time, as reported by the National Health Research 2019 [46]. In addition, it is important to consider that the comparisons between studies should contemplate the different diagnosis criteria previously defined, which may lead to results discrepancies.

Our study was developed according to the recent guidelines defined for metabolic syndrome diagnosis (IDF/NHLBI/AHA/WHF/IAS/IASO). The data collection was performed in a geographic region in Brazil where information about this topic is highly scarce. These data have huge importance since metabolic syndrome is a well-known risk factor for cardiovascular disease, type 2 diabetes, and other harmful conditions such as nonalcoholic fatty liver disease. This information underlines the urgent need for public health strategies at the Amazonas community level to prevent this major health issue.

Second, this study aimed to assess the impact of age, sex, BMI, education level, and physical activity on the likelihood that participants present metabolic syndrome. In general, our results showed that advancing age, sex (female), having a higher BMI, and a lower educational level independently increase the odds of metabolic syndrome. However, physical activity was not a significant predictor of metabolic syndrome when all those factors were considered. Although numerous studies have shown that increasing amounts of weekly physical activity have a favorable impact on metabolic syndrome [31,47], the difficulty in differentiating physical activity patterns, and a high level of sedentarism in this population can mask this relationship [48].

Independently of age, educational level, or physical activity, sex (i.e., being female) and BMI were the strongest predictors of having metabolic syndrome. In fact, in this sample, there was an association between these two factors; that is, women having significantly higher BMI values (results not shown) and a higher prevalence of abdominal obesity than men. Additionally, among a variety of possible factors to explain this sex-related increased risk, menopause has been associated with an increase in risk for several diseases such as cardiovascular disease, osteoporosis, diabetes, metabolic syndrome, and ovarian cancer [49]. Moreover, in this study, we found women are less active than men (results not shown), which is an important risk factor for obesity and metabolic syndrome. Our results corroborate that obesity may precede the development of other metabolic syndrome components [32], and interventions that address obesity and reduce waist circumference may reduce the incidence of metabolic syndrome.

Interestingly, our results indicate that independently of age, sex, BMI, or physical activity, participants with low education levels were more likely to have metabolic syndrome. According to our results, a recent meta-analysis [30], aiming to assess the association between metabolic syndrome and socioeconomic gradient, showed that the most disadvantaged people who had a lower education level, were unemployed, and of a lower income level had more vulnerability to develop metabolic syndrome. Our results support the conceptual idea that the education level of a population about the health hazards related to metabolic syndrome could be an important way to prevent this public health problem. Finally, it is well accepted that advancing age is an independent risk factor for metabolic syndrome [14]. This is explained by age-associated declines in several physiologic variables as well as unhealthy lifestyles adopted during the lifespan that substantially increases the metabolic risk factors [29]. Altogether, our results bring new insights into the Amazonas region, proposing prevention campaigns for metabolic syndrome focused on the oldest people, women, obese people, and those with a lower education level.

There were some limitations associated with this study. First of all, we acknowledge that the cross-sectional design of the present study limits conclusions regarding the direction of relationships between metabolic syndrome and modifiable explanatory factors for this phenomenon. Secondly, due to the lack of consensus about the criteria used in the definition of metabolic syndrome, as well as the huge range of ages involved, some differences when comparing to other studies should be considered. Third, although the majority of the prior studies have assessed physical activity by questionnaires, the limited ability of some participants to accurately recall past sport and leisure activities could introduce bias and lead to misclassification. The introduction of a more objective instrument to assess physical activity, i.e., accelerometers, even if in a subsample, should be considered in future studies.

To the strengths of the present study, it is important to mention that this is the first study in a large sample of adults and older adults from the state of Amazonas. In addition, clinical parameters to identify those with metabolic syndrome were assessed by specialized clinical analysis laboratory technicians. Glucose, HDL and LDL, and triglyceride levels were assessed from blood samples, which guarantee the quality and validity of the information collected.

## 5. Conclusions

This is the first study in a large sample of adults and older adults from the state of Amazonas on metabolic syndrome and their relationship with independent explanatory factors. The prevalence of metabolic syndrome found is high considering the literature. This study supports targeted interventions to face this major public health problem. In particular, blood pressure, abdominal obesity, and a lower concentration of HDL-C were the metabolic abnormalities most prevalent in this sample. This study demonstrated that advancing age, being female, having a higher BMI, and a lower educational level are key variables to be considered in public health interventions. More research in this important field of vulnerability and health research is needed specific to this region due to the divergent results from the general Brazilian population.

## Figures and Tables

**Table 1 ijerph-18-01303-t001:** Descriptive characteristics of the sample by sex.

Variables	Men (*n* = 332)			Women (*n* = 558)			*p*
	Mean	SD	CI (95%)	Mean	SD	CI (95%)	
Age (y)	61.0	20.1	60.0–64.1	59.1	19.3	59.1–62.1	0.141
SBP (mm Hg)	133.8	17.0	132.2–135.8	127.8	17.3	126.7–129.6	<0.001
DBP (mm Hg)	77.1	12.2	75.3–77.9	74.4	11.8	72.9–74.9	0.001
HR (bpm)	72.3	11.1	71.1–73.4	75.6	10.8	74.3–76.4	<0.001
GLI (mg·dL^−1^)	94.9	37.1	90.6–98.5	101.6	42.6	98.3–105.5	0.015
CHOL—total (mg·dL^−1^)	171.9	45.2	166.9–176.3	190.8	54.6	187.0–195.9	<0.001
HDL (mg·dL^−1^)	42.4	13.0	41.0–43.8	47.2	12.4	46.3–48.4	<0.001
LDL (mg·dL^−1^)	104.9	37.1	100.7–108.7	118.5	40.2	115.3–122.0	<0.001
TG (mg·dL^−1^)	149.2	106.6	132.7–152.1	150.9	92.1	140.3–154.5	0.800
WACI (cm)	89.0	12.4	88.1–90.7	85.1	11.9	84.6–86.5	<0.001
BMI (kg·m^−2^)	26.8	4.7	26.4–27.4	28.1	5.5	27.8–28.7	<0.001
PA (units)	7.9	1.1	7.8–8.0	7.5	1.1	7.5–7.7	<0.001
Education (*n*)	1.1	1.3	0.91–1.20	1.5	1.4	1.3–1.56	<0.001

Independent-samples *t*-test; MS, metabolic syndrome; SBP, systolic blood pressure; DBP, diastolic blood pressure; GLI, fasting blood glucose; CHOL, cholesterol; HDL, high-density lipoproteins; LDL, low-density lipoproteins; TG, triglycerides; WACI, waist circumference; HR, heart rate; BMI, body mass index; PA, physical activity; 95% C.I., 95% confidence interval for means.

**Table 2 ijerph-18-01303-t002:** Sex-related differences in the prevalence of each individual risk factor for metabolic syndrome.

Rsk Factors for Metabolic Syndrome	Sex				Total	
	Men		Women			
	*n* (%)	CI (%)	*n* (%)	CI (%)	*n* (%)	CI (%)
Below cut-off WACI	221 (62.8)	57.5–67.9	193 (32.7)	28.9–36.7	414 (43.9)	40.8–47.2
Above cut-off WACI	131 (37.2)	32.2–42.5	397 (67.3) **	63.3–71.1	528 (56.1)	52.8–59.3
Below cut-off TG	221 (64.2)	58.9–69.3	356 (61.9)	57.8–65.9	577 (62.8)	59.6–65.9
Above cut-off TG	123 (35.8)	32.9–43.4	219 (38.1)	65.1–72.8	342 (37.2)	34.1–40.4
Below cut-off HDL-C	163 (47.8)	43.4–53.3	199 (35.0)	31.1–39.1	362 (39.8)	36.6–43.0
Above cut-off HDL-C	178 (52.2)	46.8–57.6	370 (65.0) **	61.0–69.0	548 (60.2)	56.9–63.4
Below cut-off BP	132 (37.5)	32.4–42.8	303 (51.4)	47.2–55.5	435 (46.2)	43.0–49.4
Above cut-off BP	220 (62.5)	57.2–67.6	287 (48.6) **	44.5–52.8	507 (53.8)	50.6–57.0
Below cut-off GLI	250 (72.0)	67.0–76.7	393 (67.4)	63.4–71.2	643 (69.1)	66.1–72.1
Above cut-off GLI	97 (28.0)	23.3–33.0	190 (32.6)	28.8–36.6	287 (30.9)	27.9–33.9
Below cut-off MS	206 (60.4)	55.0–65.6	227 (47.8)	43.6–52.0	478 (52.5)	49.2–55.8
Above cut-off MS	153 (39.6)	34.4–45.0	297 (52.2) **	48.0–56.4	432 (47.5)	44.2–50.8

** Chi-squared *p* < 0.05; CI, confidence interval; WACI, waist circumference; TG, triglycerides; HDL-C, high-density lipoproteins cholesterol; BP, blood pressure; GLI, fasting blood glucose; MS, metabolic syndrome diagnosis made according to the criteria defined by the International Diabetes Federation Task Force on Epidemiology and Prevention, the National Heart, Lung, and Blood Institute, the American Heart Association, the World Heart Federation, the International Atherosclerosis Society, and the International Association for the Study of Obesity (IDF/NHLBI/AHA/WHF/IAS/IASO) [1].

**Table 3 ijerph-18-01303-t003:** Logistic regression predicting the likelihood of presenting metabolic syndrome.

Predictors	B	S.E.	Wald	df	*p*	Odds Ratio	95% C.I. for EXP(B)	
							Lower	Upper
Age (y)	0.03	0.00	32.43	1.00	<0.001	1.03	1.02	1.04
Sex (M = 1; W = 0)	−0.52	0.16	10.63	1.00	0.001	1.67	1.23	2.27
BMI (kg/m^−2^)	0.17	0.02	93.68	1.00	<0.001	1.18	1.14	1.22
Education (*n*)	−0.12	0.06	4.05	1.00	0.044	1.12	1.00	1.35
PA (units)	0.03	0.07	0.13	1.00	0.717	1.03	0.89	1.18
Constant	−6.82	0.79	74.99	1.00	<0.001	0.001		

PA, physical activity; BMI, body mass index; B, unstandardized regression weight Wald, Wald statistic test; df, degrees of freedom; 95% C.I. for EXP(B), 95% confidence interval for the odds ratio.

## Data Availability

The data presented in this study are available on request from the corresponding author.

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
