# Peer review of "Predictors of Metabolic Syndrome in Adults and Older Adults from Amazonas, Brazil"

_ijerph, 2021, doi:10.3390/ijerph18031303_

Round 1
Reviewer 1 Report
Comments:
- The background part mentioned 2 aims on line 74-76 and 103-105, I would hope to put the aims together to make it clear. Also, on line 74-76, I don’t quite understand what it means exactly to aim to “add consistency”. If it is about the underlining criteria used to define metabolic syndrome, I don’t see how this paper try to established or unified the current different criteria.
- The Amazonas is definitely unique and needs investigation in regards of metabolic syndrome, but how Amazonas are different from other populations are not well-described, and the research finding are pretty much in consensus with current findings mentioned in 88-89. The gaps and significance of the research in the manuscript are not clear.
- Line 138, if Baecke questionnaire was not originally developed for Brazilians or other populations. It would be good to address the limitation of not adapting to the local culture.
- For Table 1 and 2, the location of the stars is not clear and please change the “,” to “.” between the numbers. Also, in Table 2, although on line 163-165 chi-square was mentioned, it is confusing to read as it seems only “with risk” was compared in the statement, rather than “without risk”. Please add the 0 to the numeric values in Table 3, like 0.03 rather than .03.
- Multiple testing could be an issue if there are many predictors tested as independent ones and the authors should consider control for multiple testing using a more stringent p value cutoff.
- Line 211, a confusion matrix of the logistic regression to show classification would be appreciated.
- Line 212 – 221, the language to describe odds ratio need to be improved. For example, line 215, it should specify for each 1kg/m2 “INCREASE” in BMI, rather than the current expression. Similar issues for other sentences as well.
- Please add the details of Education and Physical activity variable in the methods section, e.g. levels of education, and what the units of physical activity like METS.
Author Response
Reviewer 1
Comments:
1. The background part mentioned 2 aims on line 74-76 and 103-105, I would hope to put the aims together to make it clear. Also, on line 74-76, I don’t quite understand what it means exactly to aim to “add consistency”. If it is about the underlining criteria used to define metabolic syndrome, I don’t see how this paper try to established or unified the current different criteria.
Response 1: To better clarify this point we have rephrased the sentence on line 74-76.
“Our study, using a reliable methodology to diagnose metabolic syndrome adds knowledge about the topic in the Amazonas region and highlights the need for strategies to prevent this emerging global health concern.”
2. The Amazonas is definitely unique and needs investigation in regards of metabolic syndrome, but how Amazonas are different from other populations are not well-described, and the research finding are pretty much in consensus with current findings mentioned in 88-89. The gaps and significance of the research in the manuscript are not clear.
Response 2: To better clarify this point, in the discussion section lines 59-66 we stated that in the last decade, several studies have contributed to estimate the prevalence of metabolic syndrome and its risk factors in many regions in Brazil. However, none of them had gathered information from the Amazonas region. From our knowledge, this is the first report about metabolic syndrome in this region. We assume that this information about Amazonas has vital importance in early risk identification and can increase the effectiveness of specific interventions aimed to prevent future deterioration of the health status. We further highlight the novelty of the present paper in the discussion.
3. Line 138, if Baecke questionnaire was not originally developed for Brazilians or other populations. It would be good to address the limitation of not adapting to the local culture.
Response 3: The reviewer is right that the Baecke questionnaire was not originally developed for Brazilians. However, this instrument has been utilized in epidemiological research worldwide and in Brazil [Gouveia et al. Arch Gerontol Geriatr 2014, 59(1),83-90; Pereira et al. Prev Med 1999, (28), 304-12; Florindo et al. J Gerontol A Biol Sci Med Sci 2002, (57), 654-7]. In addition, the validation and reliability of the Baecke questionnaire was also previously published for Brazilian population [Florindo & Latorre. Rev Bras Med Esporte, 2003 9(3),129-135]
4. For Table 1 and 2, the location of the stars is not clear and please change the “,” to “.” between the numbers. Also, in Table 2, although on line 163-165 chi-square was mentioned, it is confusing to read as it seems only “with risk” was compared in the statement, rather than “without risk”. Please add the 0 to the numeric values in Table 3, like 0.03 rather than .03.
Response 4: As suggested by the reviewer all changes were made on table 1 and table 3. We also changed the terminology for “with risk” and “without risk”.
5. Multiple testing could be an issue if there are many predictors tested as independent ones and the authors should consider control for multiple testing using a more stringent p value cutoff.
Response 5: To better clarify this point, we used direct logistic regression to assess the impact of age, sex, BMI, education, and physical activity on the likelihood that participants presented metabolic syndrome. The model tested contained modifiable (BMI, education, and physical activity) and non-modifiable variables (age and sex) that have been reported in the literature as affecting metabolic syndrome. As common in the literature for social sciences, we adopted a significance level of 5%. We further discuss this important issue.
6. Line 211, a confusion matrix of the logistic regression to show classification would be appreciated.
Response 6: As suggested by the reviewer improvements in the text has been done. To better clarify this point, this classification provides an indication of how well the model is able to predict the correct category (with metabolic syndrome/no metabolic syndrome) for each case. Our model correctly classified 71% of the cases overall (sometimes referred to as the percentage accuracy in classification).
The sentence was rephrased as following:
“The model as a whole explained between 19.7% (Cox & Snell R square) and 26.2% (Nagelkerke R square) of the variance in metabolic syndrome status, and correctly classified 71% of the cases.”
7. Line 212 – 221, the language to describe odds ratio need to be improved. For example, line 215, it should specify for each 1kg/m2 “INCREASE” in BMI, rather than the current expression. Similar issues for other sentences as well.
Response 7: As suggested by the reviewer, the language to describe odds ratio has been improved.
8. Please add the details of Education and Physical activity variable in the methods section, e.g. levels of education, and what the units of physical activity like METS.
Response 8: As suggested by the reviewer, more information about Physical activity and Education was added, as follows:
“Numerical coding for most response categories varied from 1 to 5 (Likert scale), ranging from never to always or very often. The questionnaire also provides a measure of total PA, which is the sum of these three specific domains. Numerical coding for most response categories varied from 1 to 5 (Likert scale) ranging from never to always or very often.”
“To assess the education level, participants were asked to indicate their level of schooling. The following scores were considered for this purpose: 0 = Illiterate/Incomplete primary; 1 = Complete primary/Incomplete junior high; 2 = Complete junior high/Incomplete secondary; 3 = Complete secondary/incomplete higher; 4 = Complete higher.”
Reviewer 2 Report
The aim of this paper is to look at the prevalence and risk factors associated with metabolic syndrome in the Amazonas region of Brazil. Data was collected from the Health, Lifestyle, and Functional Fitness in the Adult and Older People from Amazonas, Brazil study. Overall I found the work to be straightforward and the data convincing.
There is a lot of passive voice sentences in this paper. It would be easier to read if the writers eliminated this. Perhaps a native speaker could review the writing.
(line 55) Evidence points out the fact that individuals with metabolic syndrome are 2.5 times more likely to die from cardiovascular disease when compared with their peers with no metabolic syndrome x CITATION
(line 78) Other factors have also been shown TO PLAY a role in the development of metabolic syndromes…
Please be consistent with statistics. Custom dictates * p<0.05, ** p<0.01, *** p<0.001, **** p<0.0001
Please elaborate how low concentration of HDL-C is protective of CVD but not metabolic syndrome. Can you separate out women who had higher HDL-C and developed metabolic risk? This sentence makes it seem like the same women had low HDL-C and developed metabolic syndrome. (Line 229) Importantly, women had a higher prevalence of abdominal obesity, low HDL-C, and metabolic syndrome than men, however, opposite results were seen in men for blood pressure.
Author Response
The aim of this paper is to look at the prevalence and risk factors associated with metabolic syndrome in the Amazonas region of Brazil. Data was collected from the Health, Lifestyle, and Functional Fitness in the Adult and Older People from Amazonas, Brazil study. Overall I found the work to be straightforward and the data convincing.
We are very grateful to Reviewer 2 for this overall positive evaluation. We are very thankful for the detailed and helpful suggestions for improvement. In a thorough revision, we have now addressed all of the comments raised and feel that the manuscript has substantially been improved as a result. Our detailed responses are provided below.
- There is a lot of passive voice sentences in this paper. It would be easier to read if the writers eliminated this. Perhaps a native speaker could review the writing.
Response 1: As suggested by the reviewer, we attentively checked the manuscript for language issues and revised it accordingly.
- (line 55) Evidence points out the fact that individuals with metabolic syndrome are 2.5 times more likely to die from cardiovascular disease when compared with their peers with no metabolic syndrome (Sociedade Brasileira de Hipertensão, Sociedade Brasileira de Cardiologia, Sociedade Brasileira de Endocrinologia e Metabologia, Sociedade Brasileira de Diabetes, Sociedade Brasileira de Estudos da Obesidade, 2010).
Response 2: The citation [10] was added in the text.
- (line 78) Other factors have also been shown TO PLAY a role in the development of metabolic syndromes…
Response 3: As suggested, edits have been made in the text.
- Please be consistent with statistics. Custom dictates * p<0.05, ** p<0.01, *** p<0.001, **** p<0.0001
Response 4: As suggested, edits have been made in the text.
- Please elaborate how low concentration of HDL-C is protective of CVD but not metabolic syndrome. Can you separate out women who had higher HDL-C and developed metabolic risk? This sentence makes it seem like the same women had low HDL-C and developed metabolic syndrome.
(Line 229) Importantly, women had a higher prevalence of abdominal obesity, low HDL-C, and metabolic syndrome than men, however, opposite results were seen in men for blood pressure.
Response 5: To better clarify this point, in the discussion section we stated that, between the 5 risk factors that are considered to calculate the metabolic syndrome, the second most prevalent in both, men and women, was the low concentration of HDL-C. Several studies have been reported that a low concentration of HDL-C is an important risk factor for the development of cardiovascular diseases as well as for metabolic syndrome.
In line 229 several edits were made to clarify that men showed higher prevalence of blood pressure than women.
In this study 42.7% of women and 28.7% of men that had a low concentration of HDL-C developed metabolic syndrome risk.
Reviewer 3 Report
Manuscript Review: ijerph-1033707
Title: “Predictors of Metabolic Syndrome in an Adult Lifespan Sample from Amazonas, Brazil”
Summary
The authors carried out a cross-sectional study using data from 942 individuals participating in a local study conducted from 2016-2017 in Amazonas, Brazil. The chi-square test was used to calculate the prevalence of metabolic syndrome and logistic regression was used to estimate the effect of independent variables on metabolic syndrome. According to the authors, the prevalence of metabolic syndrome was high (47.5%). Age, sex, BMI and education were independent predictors of metabolic syndrome.
Study strengths
- Sample size and study population, which is part of a larger local study.
Points that need clarification
Title
- I suggest including the word “older” in the title, as well as in the title of the main study from which the data were extracted: “Predictors of Metabolic Syndrome in an older adult Lifespan Sample from Amazonas, Brazil”.
Introduction
- The Introduction section needs to be better structured; the authors confuse introduction with discussion.
- Page 3, lines 70-40: “It is important to underline that the lack of consensus about the criteria used in the definition of metabolic syndrome may account for some differences across studies. Another reason for study differences is the way data were collected. Some studies considered self-reported information that makes the diagnosis of metabolic syndrome less reliable.” This sentence should be part of the Discussion section, not Introduction.
- Page 3, lines 77-87: This paragraph should also be part of the Discussion.
Material and Methods
- Page 4, Sample and Study Design: Provide details of the sampling process such as inclusion and exclusion criteria and possible losses. I suggest including a flowchart of the sampling process.
- In which cities was the study conducted? Amazonas is a huge state in northwestern Brazil, covered almost entirely by the Amazon rainforest. Is the sample representative?
- Page 4, Clinical analysis: Please provide details of the biochemical analysis: kits, equipment, and analytical sensitivity for each parameter investigated.
- Page 4, lines 123-128: Provide details of how blood pressure was measured: how many readings and at which interval?
- Anthropometric Measurements: According to the authors, waist circumference was measured, but Table 2 describes abdominal obesity. How was abdominal obesity calculated? What is the reference standard used? Did the authors also measure hip circumference? This needs to be clarified.
- Page 4, Physical activity: It is important to provide details of how the physical activity score was calculated.
- Page 5: The classification of the educational level is exclusive for Brazilians. Thus, this classification should be described briefly in the Methods section.
- Page 5, Statistics: Please provide more information about how the statistical analysis was performed. Was the normality of the variables verified? Adjusted analysis? How were the variables selected?
Results:
- Page 6, Table 1: Please include definitions of all acronyms in the footnote: “FC”, “CHOL”, … Also include the statistical test used and “n (%)” for each variable. Did all variables have a normal distribution?
- Page 6, Table 1: Some variables presented in Table 1 (FC, CHOL, total, LDL) were not considered in the final model (Table 3).
- Page 7, Table 2: Abdominal obesity or “waist circumference”? Standardize the number of (*) for p-values. Table 1 and 2 are different! Also include n (%) for each variable.
- Page 8, Table 3: What does “full model” mean? Has the full model been adjusted for all variables in Table 3? If so, what was the criterion for including variables in the model? Why did the authors not consider all variables in Table 1? How was the quality of the model assessed? These characteristics must be described in the Methods section, Statistics, and in the footnote of Table 3.
Discussion
- Page 9: The information presented in the first paragraph of the discussion is correct; however, it does not include the results of Table 3. That is, the authors mention only prevalence and not predictors, as stated in the title and objectives. Therefore, Table 3 does not seem to make sense for this study.
- Line 232: “Our results support that men have a greater prevalence of hypertension in comparison to women.” Incorrect statement, the disease (hypertension) was not evaluated, only blood pressure! At most, the authors can comment on high blood pressure, but not on hypertension.
- Lines 254-264: The studies used to compare the prevalence of metabolic syndrome involved very different age groups, a fact impairing comparison. Since this study involves a specific population of older adults (mean age = 66.2 years), it is very likely that the prevalence of metabolic syndrome is higher when compared to studies on adolescents, for example.
- Line 273: The objective mentioned in line 105 was: “…and to identify independent predictors of metabolic syndrome.” It does not seem to be in line with what is mentioned in the discussion: “The second purpose of this study was to assess the impact of age, sex, BMI, education level, and physical activity on the likelihood that participants present the metabolic syndrome.”, which appears only to address sociodemographic predictors.
- Lines 282-284: Please explain the following sentence: “In fact, in this sample there was an association between these two factors, that is, women have significantly higher BMI values (results not shown) and have a higher prevalence of abdominal obesity than men”. Did the authors test the interaction between these two variables?
Conclusion
- Suggestion: Change the sentence “…on an adult lifespan sample…” to “…on an older adult and lifespan sample…” throughout the text. It needs to be clear that the participants belong to a specific age group and that they should not be compared with other age groups.
- I suggest modifying the conclusion. It seems more like a repetition of the results.
Author Response
Title: “Predictors of Metabolic Syndrome in an Adult Lifespan Sample from Amazonas, Brazil”
Summary
The authors carried out a cross-sectional study using data from 942 individuals participating in a local study conducted from 2016-2017 in Amazonas, Brazil. The chi-square test was used to calculate the prevalence of metabolic syndrome and logistic regression was used to estimate the effect of independent variables on metabolic syndrome. According to the authors, the prevalence of metabolic syndrome was high (47.5%). Age, sex, BMI and education were independent predictors of metabolic syndrome.
Study strengths
- Sample size and study population, which is part of a larger local study.
We are very grateful to Reviewer 3 for this overall positive evaluation. We are very thankful for the detailed and helpful suggestions for improvement. In a thorough revision, we have now addressed all of the comments raised and feel that the manuscript has substantially been improved as a result. Our detailed responses are provided below.
Points that need clarification
Title
- I suggest including the word “older” in the title, as well as in the title of the main study from which the data were extracted: “Predictors of Metabolic Syndrome in an older adult Lifespan Sample from Amazonas, Brazil”.
Response 1: Considering the different suggestions from reviewers 3 and 4 about the title, we revised it as follows: “Predictors of Metabolic Syndrome in Adults and Older Adults from Amazonas, Brazil”
Introduction
2. The Introduction section needs to be better structured; the authors confuse introduction with discussion.
Response 2: in order to follow the reviewer’s suggestion, substantial edits were made in the introduction.
3. Page 3, lines 70-40: “It is important to underline that the lack of consensus about the criteria used in the definition of metabolic syndrome may account for some differences across studies. Another reason for study differences is the way data were collected. Some studies considered self-reported information that makes the diagnosis of metabolic syndrome less reliable.” This sentence should be part of the Discussion section, not Introduction.
Response 3: As suggested by reviewer the sentence was deleted from the Introduction.
4. Page 3, lines 77-87: This paragraph should also be part of the Discussion.
Response 4: To better clarify this point, in this paragraph we are trying to explain why this problem is important and how does the study relate to previous work in the topic. We feel that this information is important for the reader, because it covers what we are going to explore in the results and discussion section. We also underline in this paragraph some of the major gaps in the literature that this study intends to overcome.
Material and Methods
5. Page 4, Sample and Study Design: Provide details of the sampling process such as inclusion and exclusion criteria and possible losses. I suggest including a flowchart of the sampling process.
Response 5: As suggested, more information about the sampling process were added in the methods section:
“Participants were volunteers recruited via advertisements distributed through newspapers, local radio, churches, schools and senior centers. Inclusion criteria were living in one of the three geographic areas of Brazil (Fonte Boa, Apuí, and Manaus), voluntary motivation to participate in the study, and being able to walk independently. Exclusion criteria was the inability to understand and follow the assessment protocol of the study.”
6. In which cities was the study conducted? Amazonas is a huge state in northwestern Brazil, covered almost entirely by the Amazon rainforest. Is the sample representative?
Response 6: The sample comprised participants from Fonte Boa, Apuí, and Manaus, all municipals from Amazonas, Brazil.
“For the sample size calculation, we used the GPower 3.1.9.7 software (Faul, et al. 2007). A priori, logistic regression analysis indicated that a total sample size of 824 participants was needed to achieve 85% power, odds ratio of 1.3 at the 0.05 level of significance.”
Faul, F., Erdfelder, E., Lang, A.-G., et al., (2007). G*Power 3: a flexible statistical power analysis program for the social, behavioral, and biomedical sciences. Behav. Res. Methods 39, 175–191.
7. Page 4, Clinical analysis: Please provide details of the biochemical analysis: kits, equipment, and analytical sensitivity for each parameter investigated.
Response 7: as recommended by the reviewer more detailed information about the clinical analysis was provided:
“Blood was collected by venipuncture from all participants, fasting (more than 8 hours), between 7 am and 9 am. For this purpose, needles were used, syringes and collection tubes. After collecting 7 ml of blood from the antecubital vein to a dry tube with silica gel, i.e., an accelerator of serum separation, the tubes were identified and transported to the Laboratory. After an hour of rest, separating the serum by centrifugation at room temperature for 15 minutes at 3500 rotations per minute.”
8. Page 4, lines 123-128: Provide details of how blood pressure was measured: how many readings and at which interval?
Response 8: As suggested by the reviewer, more information was added, as follows:
At least two measurements were made (separate repeated measurements by 1–2 min.), taking the average value between both as long as the difference did not exceed 5 mm/Hg.
9. Anthropometric Measurements: According to the authors, waist circumference was measured, but Table 2 describes abdominal obesity. How was abdominal obesity calculated? What is the reference standard used? Did the authors also measure hip circumference? This needs to be clarified.
Response 9: to clarify this point, waist circumference was used as an abdominal obesity indicator. We followed the consensus of theIDF/NHLBI/AHA/WHF/IAS/IASO (Alberti et al., 2009) to classify the abdominal obesity, as follow: circumference ³ 94 cm in men and ³ 80 cm in women. In this study we did not consider hip circumference nor any ratio.
10. Page 4, Physical activity: It is important to provide details of how the physical activity score was calculated.
Response 10: This point was already answered to reviewer 1 in response 8.
11. Page 5: The classification of the educational level is exclusive for Brazilians. Thus, this classification should be described briefly in the Methods section.
Response 11: This point was already answered to reviewer 1 in response 8.
12. Page 5, Statistics: Please provide more information about how the statistical analysis was performed. Was the normality of the variables verified? Adjusted analysis? How were the variables selected?
Response 12: to better clarify the Statistics analyses, several edits were made in the Statistics section.
Results:
13. Page 6, Table 1: Please include definitions of all acronyms in the footnote: “FC”, “CHOL”, … Also include the statistical test used and “n (%)” for each variable. Did all variables have a normal distribution?
Response 13: As suggested by the reviewer, several edits were made in table 1. As answered before (response 12), detailed information about the statistical analyses were added in the text.
14. Page 6, Table 1: Some variables presented in Table 1 (FC, CHOL, total, LDL) were not considered in the final model (Table 3).
Response 14: As explained in the statistical analyses section, several models were tested in order to identify what factors better explained the likelihood that participants would present metabolic syndrome. The final model contained age, sex (non-modifiable variables), and BMI, education, and physical activity (modifiable variables).
15. Page 7, Table 2: Abdominal obesity or “waist circumference”? Standardize the number of (*) for p-values. Table 1 and 2 are different! Also include n (%) for each variable.
Response 15: As clarified in response 5, waist circumference was used as an abdominal obesity indicator circumference ³ 94 cm in men and ³ 80 cm in women. In table 2, we depict the participants with/without risk for abdominal obesity.
16. Page 8, Table 3: What does “full model” mean? Has the full model been adjusted for all variables in Table 3? If so, what was the criterion for including variables in the model? Why did the authors not consider all variables in Table 1? How was the quality of the model assessed? These characteristics must be described in the Methods section, Statistics, and in the footnote of Table 3.
Response 16: to better clarify this point, logistic regression allows to assess how well a set of predictors variables (in our case, age, sex, BMI, education, and physical activity) predicts or explains one categorial variable (0 = no presence of metabolic syndrome; 1 = with metabolic syndrome). This analysis also gives us an indication of the adequacy of the full model (all predictors together). Finally, it also provides us an indication of the relative importance of each predictor variable or the interaction among predictors variables.
As explained in response 14, several models were tested in order to identify what factors better explained the likelihood that participants would present metabolic syndrome. The quality of the models were assessed by Cox & Snell R Square and Nagelkerk R Square values, that provide an indication of the amount of variation in the syndrome metabolic status explained by the model. Several edits were made in the methods section in order to clarify all this issues.
Discussion
17. Page 9: The information presented in the first paragraph of the discussion is correct; however, it does not include the results of Table 3. That is, the authors mention only prevalence and not predictors, as stated in the title and objectives. Therefore, Table 3 does not seem to make sense for this study.
Response 17: To better clarify this point, we have added more information about the second purpose of this study. The main results are now presented in the first paragraph.
18. Line 232: “Our results support that men have a greater prevalence of hypertension in comparison to women.” Incorrect statement, the disease (hypertension) was not evaluated, only blood pressure! At most, the authors can comment on high blood pressure, but not on hypertension.
Response 18: we agree with the reviewer, several edits were made across the text.
19. Lines 254-264: The studies used to compare the prevalence of metabolic syndrome involved very different age groups, a fact impairing comparison. Since this study involves a specific population of older adults (mean age = 66.2 years), it is very likely that the prevalence of metabolic syndrome is higher when compared to studies on adolescents, for example.
Response 19: We agree with the reviewer. In the limitations section of this study, we approach this issue, as follows: “Secondly, due to the lack of consensus about the criteria used in the definition of metabolic syndrome, as well as to the huge range of ages involved, some differences when comparing to other studies should be considered.”
20. Line 273: The objective mentioned in line 105 was: “…and to identify independent predictors of metabolic syndrome.” It does not seem to be in line with what is mentioned in the discussion: “The second purpose of this study was to assess the impact of age, sex, BMI, education level, and physical activity on the likelihood that participants present the metabolic syndrome.”, which appears only to address sociodemographic predictors.
Response 20: to better clarify this point, we have added more information about the second objective.
21. Lines 282-284: Please explain the following sentence: “In fact, in this sample there was an association between these two factors, that is, women have significantly higher BMI values (results not shown) and have a higher prevalence of abdominal obesity than men”. Did the authors test the interaction between these two variables?
Response 21: Too better explain this sentence, we did not test the interaction between these two variables, however, we tested if there was a significant difference in the mean of BMI and waist circumference for men and women. For both variables, there was a significant difference, with women showing higher values than men.
Conclusion
22. Suggestion: Change the sentence “…on an adult lifespan sample…” to “…on an older adult and lifespan sample…” throughout the text. It needs to be clear that the participants belong to a specific age group and that they should not be compared with other age groups.
Response 22: Considering the different suggestions from reviewers 3 and 4 about the expressions used, we have revised this sentence throughout the text as follows: “in adults and older adults”
23. I suggest modifying the conclusion. It seems more like a repetition of the results.
Response 23: Following the reviewer’s suggestion, several changes were made in the discussion.
Reviewer 4 Report
This is an appropriate evaluation of the likelihood of MetS in an understudied area of Brazil while looking at sex comparisons and predictors.
Elements of this paper can be improved to be in line with other papers in this field.
Consider revising the title to just state adults
Line 41: “overall prevalence of metabolic syndrome was 47.5%, which is relatively high.” I would refrain from making statements such as relatively high unless you have a referenced statistical comparison
Line 44: “These predictors need to be targeted in individually-tailored interventions in the Amazonas region.” Consider revising the conclusion to perhaps include the need for early intervention due to association with future health risk
Line 52: “high blood sugar” more commonly stated as elevated blood glucose
Line 55: “Evidence points out the fact” this is not needed
Line 74: “Our study aims to add consistency to the present evidence on the topic of metabolic syndromes and highlights the urgent need for strategies to prevent these emerging global health concerns also in the Amazonas region.” Perhaps you can define for the reader what is specifically different about the Amazonas region that would be divergent from Brazil
Line 169: more commonly expressed as p<0.05
Table 1: I am little confused about the decision to do this table. It would be expected that the risk factors would be higher among those with metabolic syndrome as they are used in the classification. A sex based comparison seems more appropriate for a descriptive table here.
Table 2: I would consider using different terminology that without risk and with risk. Perhaps below or above the cut off listed for that risk factor
Line 203: However, men had higher prevalence in Blood pressure [c2 203 (1, n= 942) = 17.03;
I would state elevated blood pressure
Line 245:
“Regarding central obesity, our results corroborate that women have a higher prevalence of abdominal 244 obesity than men [37]. This might be explained by the more favorable fat distribution in women. Since 245 abdominal obesity is highly correlated with metabolic diseases, efforts to reduce or prevent the 246 deposition of intra-abdominal body fat might serve to reduce or prevent metabolic syndrome, 247 particularly in women.”
Is this based on age? Women tend to have more central obesity after menopause but lower levels than men prior to menopause.
Given the differences in your rates of metabolic syndrome than other studies in Brazil can you provide a hypothesis as to why? Are education levels lower than the rest of Brazil? Differences in diet and PA? these are stark differences within the same country and so should be addressed. For instance in Canada it would not be unusual to see major differences in metabolic syndrome between a major city and a reserve in the north but that would need to be addressed in the comparison as the context is very different.
Line 280: You state that PA isn’t a contributor to MetS in your population. I would like to see how your PA values compare to what is seen in other studies since you mention that it could be due to high levels of sedentarism. Also consider whether the PA tool you used provided adequate data.
I think you need to have a statement in your conclusions that more research is needed specific to this region due to the highly divergent results from the general Brazilian population.
Author Response
This is an appropriate evaluation of the likelihood of MetS in an understudied area of Brazil while looking at sex comparisons and predictors.
Elements of this paper can be improved to be in line with other papers in this field.
We are very grateful to Reviewer 4 for this overall positive evaluation. We are very thankful for the detailed and helpful suggestions for improvement. In a thorough revision, we have now addressed all of the comments raised and feel that the manuscript has substantially been improved as a result. Our detailed responses are provided below.
- Consider revising the title to just state adults
Response 1: we have accepted this suggestion. However, as suggested by reviewer 1, the word older adults also represents very well our sample. The new title is: “Predictors of Metabolic Syndrome in Adults and Older Adults from Amazonas, Brazil”
- Line 41: “overall prevalence of metabolic syndrome was 47.5%, which is relatively high.” I would refrain from making statements such as relatively high unless you have a referenced statistical comparison
Response 2: we agree with the reviewer. The sentence was revised as follows: “The overall prevalence of metabolic syndrome was 47.5%.”
- Line 44: “These predictors need to be targeted in individually-tailored interventions in the Amazonas region.” Consider revising the conclusion to perhaps include the need for early intervention due to association with future health risk
Response 3: As suggested by the reviewer, the conclusion was rephrased as follows: “Due to the association of metabolic syndrome with deterioration of the health status, this study sustain the need of early public health interventions in the Amazonas region”
- Line 52: “high blood sugar” more commonly stated as elevated blood glucose
Response 4: As suggested by the reviewer, the text was edited.
- Line 55: “Evidence points out the fact” this is not needed
Response 5: As suggested by the reviewer, the text was edited.
- Line 74: “Our study aims to add consistency to the present evidence on the topic of metabolic syndromes and highlights the urgent need for strategies to prevent these emerging global health concerns also in the Amazonas region.” Perhaps you can define for the reader what is specifically different about the Amazonas region that would be divergent from Brazil
Response 6: As suggested by the reviewer, we added more specific information about the highest socioeconomic and health inequalities seen in the State of Amazonas, that puts serious challenges to the health public system. The following sentences were added: “Our study, using a reliable methodology to diagnose metabolic syndrome, adds knowledge about the topic in the Amazonas region where the information is scarce. To the best of our knowledge, this is the first study in a large adult lifespan sample from this region of Brazil. This issue reaches a greater importance in this particular region, because it is characterized by socioeconomic and health inequalities, that puts serious challenges to the health public system (Garnelo et al., 2017)”.
- Line 169: more commonly expressed as p<0.05
Response 7: As suggested by the reviewer the text was edited.
- Table 1: I am little confused about the decision to do this table. It would be expected that the risk factors would be higher among those with metabolic syndrome as they are used in the classification. A sex based comparison seems more appropriate for a descriptive table here.
Response 8: As suggested by the reviewer, table 1 was revised accordingly.
- Table 2: I would consider using different terminology that without risk and with risk. Perhaps below or above the cut off listed for that risk factor
Response 9: As suggested by the reviewer the text was edited.
- Line 203: However, men had higher prevalence in Blood pressure [c2 203 (1, n= 942) = 17.03;
I would state elevated blood pressure
Response 10: As suggested by the reviewer the text was edited.
- Line 245: “Regarding central obesity, our results corroborate that women have a higher prevalence of abdominal obesity than men [37]. This might be explained by the more favorable fat distribution in women. Since abdominal obesity is highly correlated with metabolic diseases, efforts to reduce or prevent the deposition of intra-abdominal body fat might serve to reduce or prevent metabolic syndrome, particularly in women.” Is this based on age? Women tend to have more central obesity after menopause but lower levels than men prior to menopause.
Response 11: As described in the introduction, this study had two main purposes: (1) to estimate the prevalence of the individual and general components of metabolic syndrome, considering sex-related differences, and (2) to assess the impact of age, sex, BMI, education level, and physical activity on the likelihood that participants present the metabolic syndrome. In the first analyses we used independent-samples t-test and Chi-square test for independent to compare means and proportions, and age was not considered on this analysis. For the second purpose, in a life span perspective we considered the individual contribution of age to explain the variance on metabolic syndrome status. In addition, odds ratio was calculated in order to quantify the change in odds of having metabolic syndrome when age increases by one year.
- Given the differences in your rates of metabolic syndrome than other studies in Brazil can you provide a hypothesis as to why? Are education levels lower than the rest of Brazil? Differences in diet and PA? these are stark differences within the same country and so should be addressed. For instance in Canada it would not be unusual to see major differences in metabolic syndrome between a major city and a reserve in the north but that would need to be addressed in the comparison as the context is very different.
Response 12: to better explain the differences in metabolic syndrome between population from regions of Brazil we added the following sentence in the discussion section: “This can be explained in part for the socioeconomic and health inequalities that puts serious challenges to the health public system (Garnelo et al., 2017), as well as for a lower proportion of people practicing exercise in leisure time, as reported by the Health National Research 2019 (Ministério da Saúde, 2020).”
- Line 280: You state that PA isn’t a contributor to MetS in your population. I would like to see how your PA values compare to what is seen in other studies since you mention that it could be due to high levels of sedentarism. Also consider whether the PA tool you used provided adequate data.
Response 13: As answered to the prior question, the Health National Research 2019 report published in 2020, mentioned that the regions North of Brazil have presented lower proportion of people practicing exercise in leisure time in comparisons to other regions. In this study physical activity was assessed by a questionnaire that can introduce bias and lead to misclassification because the limited ability of some participants to accurately recall past sport and leisure activities. We have acknowledged this limitation in the discussion section.
- I think you need to have a statement in your conclusions that more research is needed specific to this region due to the highly divergent results from the general Brazilian population.
Response 14: the conclusion of this study was rephrased as suggested by the reviewer.
Round 2
Reviewer 1 Report
Thanks for addressing the comments.